# MIDTRAINING BRIDGES PRETRAINING AND POSTTRAINING DISTRIBUTIONS

## ABSTRACT

Recently, many language models have been pretrained with a "midtraining" phase, in which higher quality, often instruction-formatted data, is mixed in at the end of pretraining. Despite the popularity of this practice, there is little scientific understanding of this phase of model training or why it is effective. In this work, we conduct the first systematic investigation of midtraining through controlled experiments with language models pretrained from scratch and fine-tuned on supervised finetuning datasets in different domains. We find that when compared after supervised fine-tuning, the effectiveness of midtraining is highest in the math and code domains, where midtraining can best reduce the syntactic gap between pretraining and posttraining data. In these cases, midtraining consistently outperforms continued pretraining in both in-domain validation loss as well as pretraining data forgetting after posttraining. We conduct ablations on the starting time of the midtraining phase and mixture weights of the midtraining data, using code midtraining as a case study, and find that timing has a greater impact than mixture weights, with earlier introduction of specialized data, yielding greater benefits indomain as well as preserving general language modeling better. These findings establish midtraining as a domain adaptation technique that compared to continued pretraining yields better performance through reduced forgetting.[1]

## 1 INTRODUCTION

The success of large language models has mostly been driven by scaling model and data size. Though many interventions seem promising, they may wash out at scale. Therefore, when methodological interventions are simple yet widely adopted across model scales, they merit attention. One such intervention is *midtraining*: breaking pretraining into two or more stages in which the latter stages incorporate higher-quality data from specialized domains such as mathematics and coding, as well as instruction-formatted data (Hu et al., 2024b; Dubey et al., 2024; OLMo Team et al., 2025). While this approach has shown improved performance after posttraining (Hu et al., 2024b), there is surprisingly little systematic study of when and why midtraining works (Wang et al., 2025).

This raises key questions for practitioners: When is midtraining expected to work better than simply posttraining a model, or doing continued pretraining on a target domain? What types of data are most effective in a midtraining data mixture? Finally, how does midtraining compare to continued pretraining in specialized domains? To address these questions, we conduct the first systematic investigation of midtraining through controlled experiments on models pretrained from scratch with different midtraining settings. We systematically vary the type of midtraining data, the timing of its introduction and mixture weights, followed by supervised fine-tuning on diverse datasets. **Our investigation reveals three key findings:**

- Midtraining is highly effective at improving downstream loss on a few key domains such as math and code. In particular, it is effective in improving accuracy on domains that are farther from the general pretraining distribution as represented by web text.

- Midtraining reduces catastrophic forgetting compared to both direct fine-tuning of a pretrained model as well as continued pretraining given the same number of tokens.

---

[1]Data and code are available at `https://anonymous.4open.science/r/all_in_one_pretraining-0EF9/`.

- The timing of introducing midtraining data into the training process is important, more so than the mixture weight of the midtraining data source.

Put together, these findings cast a new light on our understanding of midtraining's role in the LM training process – it can be viewed as effectively "bridging" the distribution gap between pretraining and out-of-domain fine-tuning data, introducing a smooth transition between the two at an appropriate time in the training process.

## 2 PRELIMINARIES

In this section, we define what we refer to as midtraining throughout this paper. While this term has been used colloquially by model developers, it lacks a standard definition, so we establish our working definition for clarity.

### 2.1 TRAINING SEQUENCE DEFINITIONS

Modern language model training typically follows a sequential approach where models are trained on different data distributions in a specific order. Rather than attempting to define training phases by their data characteristics or objectives, which often overlap in practice, we adopt a sequence-based framework that defines phases by their order in the training process.

Let $\theta \in \Theta$ denote model parameters and $\mathcal{X}$, $\mathcal{Y}$ be input and output spaces. A **training phase** is a tuple $(D, \mathcal{L}, n)$ where $D$ is a distinct data distribution over $\mathcal{X} \times \mathcal{Y}$, $\mathcal{L} : \Theta \times (\mathcal{X} \times \mathcal{Y}) \to \mathbb{R}$ is a loss function, and $n \in \mathbb{N}^+$ is the number of training steps taken in that phase. A **training sequence** is an ordered collection $S = \{(D_i, \mathcal{L}_i, n_i)\}_{i=0}^N$ of training phases, where the parameters $\theta_i$ resulting from training up to phase $i$ serve as the initialization for phase $i + 1$.

**Pretraining** is the initial training phase $(D_0, \mathcal{L}_0, n_0)$ in the sequence. In practice, pretraining often uses large, diverse sources of data, such as from the web. The objective for decoder-based language models, which we examine, is typically next-token prediction, $\mathcal{L}_0(\theta; x) = -\sum_{t=1}^{|x|} \log p_\theta(x_t | x_{<t})$. Pretraining typically consumes the majority of steps taken during training, so that $\forall i > 0, n_0 >>> n_i$.

**Midtraining** refers to intermediate training phases $(D_i, \mathcal{L}_i, n_i)$ where $0 < i < N$. Midtraining data is typically more specialized than general pretraining data, often including domain-specific content (code, math) and instruction-formatted data, while maintaining a mixture with general pretraining data. However, like pretraining, midtraining updates all model parameters. Typically, midtraining is a longer phase compared to finetuning, but shorter than the preceding pretraining phase, $n_0 > n_i > n_N$. There can potentially be multiple midtraining phases as well in the case of multi-stage pretraining curricula, but we focus on one-stage midtraining in this paper.

**Finetuning** is the final training phase $(D_N, \mathcal{L}_N, n_N)$ in the sequence. Finetuning usually uses task-specific datasets and may employ diverse objectives or preference-based methods rather than next-token prediction. Typically, finetuning uses the least number of steps, although this is not necessarily always the case.

Note that our framework reflects current popular practices for language model training, rather than prescribing an optimal training methodology. Additional edge cases include multi-stage finetuning, cyclic training schedules, and mixture schedules where distribution weights vary continuously.

### 2.2 RELATIONSHIP TO CURRICULUM LEARNING AND CONTINUED PRETRAINING

**Curriculum learning** The original definition of curriculum learning focused on gradually increasing the difficulty or diversity of training examples throughout the course of training (Elman, 1993; Bengio et al., 2009). However, this term has evolved to generally mean any strategic ordering of training data (Soviany et al., 2022). In this general view, it encompasses any training strategy in which the sequence $S = \{(D_i, \mathcal{L}_i, n_i)\}_{i=0}^N$ is designed according to some principled ordering criterion—whether based on example difficulty, domain progression, task complexity, data quality, or other strategic considerations. Therefore, midtraining can be viewed as a type of curriculum learning that operates at a distributional level rather than over individual examples: instead of ordering

specific training instances by difficulty or other criteria, midtraining strategically organizes different data distributions or domain mixtures across discrete training phases. This coarse-grained approach focuses on when to introduce entire types of data during pretraining rather than sequencing of individual examples.

**Continued pretraining** Continued pretraining typically refers to additional pretraining of an already-pretrained model on domain-specific data, where the data distribution shifts completely to the target domain (Gururangan et al., 2020; Beltagy et al., 2020). This approach commits entirely to domain-specific data after a certain point in training, potentially losing general capabilities. In contrast, midtraining maintains mixed distributions throughout the intermediate phase, preserving general pretraining data alongside specialized data. Formally, continued pretraining can be viewed as the limiting case where the midtraining distribution $D_i$ consists entirely of domain-specific data (0% original pretraining data, 100% domain data), though our results suggest this pure approach may be suboptimal. Implementation details such as optimizer state handling and learning rate schedule may vary, but most often midtraining continues on the original pretraining schedule, preserving optimizer state, while continued pretraining often operates on a new schedule.

## 3 Experimental Setting

### 3.1 Training Setup

**Pretraining** We pretrain models from the Pythia family ranging in size from 70M-410M parameters on C4 web data (Raffel et al., 2020; Biderman et al., 2023). In all cases, we train for 128B tokens (approx. 61k steps) with a cosine learning rate schedule with a maximum learning rate of 3e-4 and the AdamW optimizer (Loshchilov & Hutter, 2019). We chose to fix the training budget at a point past Chinchilla-optimality for all models (Hoffmann et al., 2022), in order to ensure that models have stabilized by the point at which midtraining data has been introduced, at least for later insertion points of midtraining data. We describe the exact training setup in Appendix A.

**Midtraining** We use five midtraining mixtures spanning popular domains: code (Starcoder), math, instructions (FLAN), general knowledge/QA, and high-quality web data (DCLM). Table 1 details each mixture's composition and sources. All mixtures are introduced at varying start points (Starcoder: 6k steps, Math: 20k steps, others: 40k steps) based on data availability to prevent repetition. We compare against a control condition continuing C4 pretraining for the same number of tokens, keeping all other training details identical.

STARCODER (CODE) Our code mixture is a subset of the Starcoder pretraining dataset (Li et al., 2023), which contains code in many languages. Note that we use code from all languages, rather than Python.

| Midtrain mix | Num. Tokens (B) | Sources |
|---|---|---|
| Starcoder | 196 | (Li et al., 2023) |
| Math | 12 | (Yue et al., 2023; Toshniwal et al., 2024) |
| FLAN | 3.5 | (Wei et al., 2022) |
| KnowledgeQA | 9.6 | (Hu et al., 2024a) |
| DCLM | 51 | (Li et al., 2024b) |

Table 1: Midtraining mixes used in our experiments and dataset(s) from which they were derived.

MATH The math mixture combines mathematical reasoning problems from the MAmmoTH (Yue et al., 2023) and OpenMathInstruct (Toshniwal et al., 2024) datasets, featuring step-by-step explanations.

FLAN (INSTRUCTIONS) Our instruction-formatted data comes from a processed version of the FLAN collection, which includes diverse task instructions and responses across natural language tasks (Wei et al., 2022).

KNOWLEDGEQA (GENERAL KNOWLEDGE AND QA) The KnowledgeQA mixture is taken from Hu et al. (2024b)'s midtraining mix, and focuses on general knowledge and dialogue. However, to distinguish the midtraining mixes further, the StackOverflow portion of this dataset is removed.

DCLM (HIGH-QUALITY WEB) Our high-quality web data is a subset of the DCLM pretraining dataset, representing web content with improved quality filtering compared to C4 (Li et al., 2024b).

**Downstream Evaluation**  We fine-tune models on the datasets GSM8k (Cobbe et al., 2021), SciQ (Welbl et al., 2017), CodeSearchNet-Python (Husain et al., 2019), and LIMA (Zhou et al., 2023) – chosen to span the domains covered by our midtraining mixtures. This allows us to test cases where the midtraining mixture is aligned or misaligned with the SFT dataset. We used standard language model supervised fine-tuning for all datasets. For information on the posttraining setup, see Appendix B.

**Catastrophic Forgetting Evaluation**  A key concern with supervised fine-tuning is whether introducing specialized data causes models to forget general capabilities acquired during pretraining. We measure catastrophic forgetting by evaluating cross-entropy loss on the original pretraining distribution by measuring loss on held-out C4 data. This approach follows established practices for measuring forgetting in language models (Luo et al., 2024; Kemker et al., 2018; Li et al., 2024a).

# 4 WHICH DOWNSTREAM TASKS BENEFIT MOST FROM MIDTRAINING?

To identify which tasks benefit most from midtraining, we evaluate all midtraining mix and SFT dataset combinations. For each combination, we measure validation loss on the target domain (adaptation effectiveness) and C4 validation loss after fine-tuning (forgetting). We average results over 5 seeds after hyperparameter search for each checkpoint.

Midtraining benefits are highly domain-specific: specialization on code yields the largest gains on code tasks, while math-focused midtraining helps mathematical-reasoning tasks. Mismatched midtraining provides minimal benefit, and general instruction mixes (e.g., FLAN) produce little improvement. Full per-dataset results and numerical comparisons are reported in Table 2 and Appendix D.

Table 2: SFT and C4 validation losses for the 410M model across downstream datasets and midtraining mixtures (5 seeds per SFT dataset). Bold indicates best within each dataset. Percentages denote the specialized data proportion mixed with C4 during midtraining.

| Dataset | Midtrain Mix | Validation Loss | |
|---|---|---|---|
| | | SFT | C4 |
| **Pycode** | | | |
| | C4 | 2.264 | 5.058 |
| | Starcoder (20%) | **2.091** | **4.611** |
| | Math (12%) | 2.274 | 5.096 |
| | FLAN (5%) | 2.252 | 5.047 |
| | KnowledgeQA (20%) | 2.264 | 5.094 |
| | DCLM (20%) | 2.262 | 5.157 |
| **GSM8K** | | | |
| | C4 | 0.956 | 4.937 |
| | Starcoder (20%) | 0.944 | 4.951 |
| | Math (12%) | **0.918** | **5.038** |
| | FLAN (5%) | 0.958 | 4.970 |
| | KnowledgeQA (20%) | 0.961 | 4.966 |
| | DCLM (20%) | 0.949 | 4.966 |
| **LIMA** | | | |
| | C4 | 3.490 | 3.166 |
| | Starcoder (20%) | **3.422** | **3.155** |
| | Math (12%) | 3.507 | 3.168 |
| | FLAN (5%) | 3.513 | 3.164 |
| | KnowledgeQA (20%) | 3.501 | 3.163 |
| | DCLM (20%) | 3.496 | 3.158 |
| **SciQ** | | | |
| | C4 | 3.271 | 6.873 |
| | Starcoder (20%) | 3.261 | 6.831 |
| | Math (12%) | **3.257** | 6.894 |
| | FLAN (5%) | 3.277 | **6.771** |
| | KnowledgeQA (20%) | 3.273 | 6.814 |
| | DCLM (20%) | 3.270 | 6.874 |

## 5   WHAT DATA IS MOST EFFECTIVE FOR MIDTRAINING?

Having established that midtraining effects are domain-specific, we now ask: *what determines the strength of these domain-specific effects*? Across midtraining-target pairs, we see improvements ranging from negligible (e.g. FLAN → coding) to strong (e.g. Starcoder → coding).

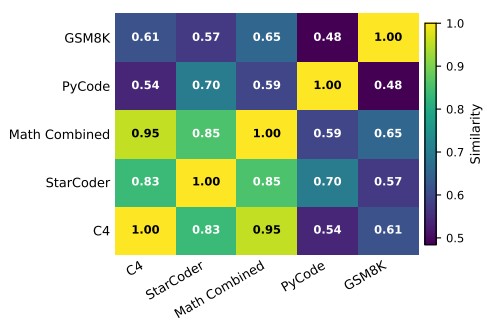

Figure 1: Example similarity matrix between pre/midtrain and posttraining datasets. For the complete matrix, see Appendix C.

Although the pairs that work the best are intuitive, we ask whether we can quantify this intuition through measurable proximity between data distributions.

To test whether optimal midtraining data "bridges" the gap between pretraining and target distributions, we simply use token-distributional similarity between datasets, which we discuss in more detail in Appendix C. Figure 1 shows a reduced subset as an example: we can see that C4 and Pycode are fairly dissimilar (0.54 similarity), but the blend containing 29% Starcoder data brings this closer to Pycode (0.7 similarity).

> **Finding 1.** Midtraining benefits are highly domain-specific. Code-focused midtraining (Starcoder) yields large gains on coding tasks, and math-focused midtraining improves mathematical reasoning. Mismatched domains provide little benefit, and broad instruction data (FLAN) also shows minimal effect.

### 5.1   PROXIMITY AND BRIDGING EFFECTS

To understand why certain midtraining mixtures are effective, we test the hypothesis that optimal midtraining data "bridges" the gap between pretraining (C4) and target SFT datasets. Using the simple token similarity metric, we measure the "proximity advantage" of each midtraining mixture—how much closer it brings the model to the target posttraining distribution compared to continuing with C4 alone. Results are shown in Figure 2.

We find a clear relationship between proximity advantage and downstream performance improvements across model sizes. The correlations are particularly strong for smaller models ($r = 0.869$, $p < 0.001$ for 70m), suggesting that effective midtraining data serves as a distributional stepping stone from general pretraining to specialized target domains. This bridging effect appears to be most beneficial when the gap between pretraining and target distributions is large, consistent with our hypothesis that midtraining helps models adapt gradually rather than requiring abrupt distributional shifts during fine-tuning.

> **Finding 2.** Effective midtraining data acts as a distributional bridge between pretraining and target datasets when considering token patterns.

### 5.2   MIDTRAINING VS. CONTINUED PRETRAINING

Our results so far suggest that effective midtraining data serves as a bridge between general pretraining and specialized posttraining data. However, a question that follows is why midtraining is necessary: continued pretraining on domain-specific data also aims to adapt the model toward a target domain. Why not simply pretrain normally and then switch to domain-specific data entirely?

To examine this, we compare the effect of midtraining with continued pretraining in which the mixture weight switches to 100% specialized data. For code, we compare the default Starcoder midtraining mix (20% mixture weight, starting from 12.6B tokens) with 100% Starcoder data starting

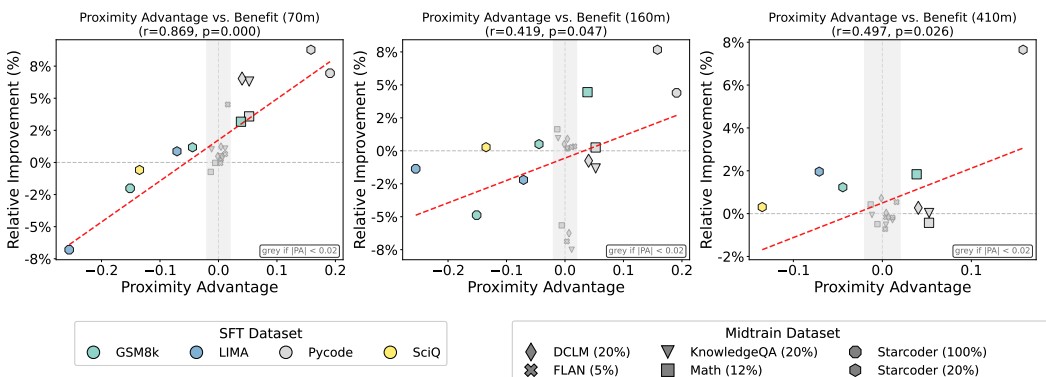

Figure 2: Relationship between proximity advantage and midtraining performance improvements for pairs of midtraining and SFT datasets. Each data point represents a (midtrain, SFT) pair, where the color indicates the SFT dataset and shape represents midtrain dataset. Proximity advantage (dist(C4, SFT) - dist(midtrain, SFT)) indicates how much closer midtraining data brings the model to the target SFT dataset compared to the base pretraining data. Proximity advantage pairs near zero are greyed out for clarity but included in calculations. Relative improvement is measured against the base model pretrained on C4.

from 83B tokens. For math, we compare the math midtraining mix with 100% math data starting from 105B tokens.[2]

Results in Table 3 show that midtraining consistently outperforms continued pretraining across both domains and model sizes for both in-domain performance and C4 retention after fine-tuning. As this pattern holds for both code and math domains, this suggests that maintaining some general pretraining data is useful during domain adaptation, even for models specialized for a specific domain. This supports our intuition gained from prior sections that domain adaptation benefits from gradual distributional shifts at the token level rather than abrupt changes.

Table 3: SFT and C4 validation losses for 70M and 160M models comparing default midtraining mixes to continued pretraining on only the midtraining data (100%), averaged across 5 seeds. Bold indicates best performance within each dataset/model combination.

| Model | Dataset | Mix | SFT | C4 |
|---|---|---|---|---|
| **70M** | | | | |
| | Pycode | Pretrain-only | 2.633 | 6.067 |
| | | Starcoder (20%) | **2.530** | **5.996** |
| | | Ctd. pretrain (Starcoder) | 2.580 | 6.160 |
| | GSM8K | Pretrain-only | 1.405 | 6.409 |
| | | Math (12%) | **1.359** | **6.391** |
| | | Ctd. pretrain (Math) | 1.401 | 6.422 |
| **160M** | | | | |
| | Pycode | Pretrain-only | 2.382 | 5.247 |
| | | Starcoder (20%) | **2.205** | **5.104** |
| | | Ctd. pretrain (Starcoder) | 2.283 | 5.373 |
| | GSM8K | Pretrain-only | 1.188 | 5.339 |
| | | Math (12%) | **1.139** | **5.198** |
| | | Ctd. pretrain (Math) | 1.180 | 5.364 |

**Finding 3.** Maintaining a mixture with general data in midtraining is preferable to continued pretraining on specialized data alone.

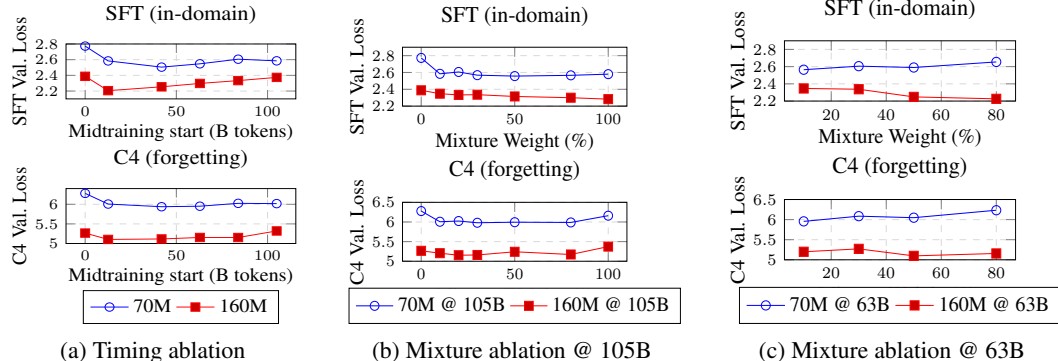

(a) Timing ablation     (b) Mixture ablation @ 105B     (c) Mixture ablation @ 63B

Figure 3: Pycode ablations. (A) Timing effect (fixed 20%). (B) Mixture effect at 105B. (C) Mixture effect at 63B. Earlier start increases sensitivity to mixture weight, especially for 70M; 160M is more robust. Top: SFT after Pycode finetuning. Bottom: C4 losses (forgetting).

## 6 WHEN AND HOW MUCH MIDTRAINING DATA SHOULD BE INTRODUCED?

Given that a midtraining dataset's proximity to the finetuning dataset matters, and that we want to balance midtraining data with general pretraining data, we next investigate when to integrate midtraining data and what mixture weight to use. Using the Starcoder midtraining mixture, we vary both timing (10%-80% through pretraining) and mixture weights (10%-80% of training data). When testing the effect of timing, we use a fixed mixture weight of 20% for Starcoder and vary the time at which the midtraining phase begins. When testing the effect of mixture weight, we fix the step at which midtraining starts at two specific points in training: 63B and 105B tokens, while varying the mixture weight of Starcoder as compared to C4.

When varying mixture proportions from a fixed starting point, effects were modest compared to varying the timing of introduction across different pretraining stages at a fixed 20% mixture weight (Figure 3(a)). The timing effects are substantial: for the 160m model, early introduction at 12B tokens achieves 2.205 validation loss compared to 2.374 when delayed to 105B steps. Similarly, the 70m model performs best with introduction at 42B steps (2.505) versus later introduction. However, we note that timing and mixture weight also interact, and mixture weight appears to be more impactful when midtraining data is introduced earlier, as the range of best and worst mixture weights roughly doubles at 63B tokens compared to at 105B tokens. This suggests that when to introduce specialized data may be more critical than how much specialized data to include, at least within our experimental ranges. See Figure 3(b–c) for mixture weight comparisons.

Relatedly, Figure 4 illustrates how midtraining benefits evolve over the course of pretraining for the 20% Starcoder mix (160m model). We finetune checkpoints from different pretraining steps on Pycode and measure both in-domain and C4 validation loss after fine-tuning. In-domain advantages emerge quickly after midtraining introduction (6k steps), while the C4 retention benefits develop more gradually, becoming apparent after approximately 20k steps. This temporal pattern suggests that early introduction of specialized data provides sufficient time for both immediate domain adaptation and gradual integration with general capabilities.

> **Finding 4.** The timing of midtraining is more critical than the mixture weight; early introduction of specialized data in the code domain leads to stronger in-domain gains and better retention of general capabilities.

## 7 HOW DOES MIDTRAINING CHANGE MODEL REPRESENTATIONS?

In the previous sections, we have established the midtraining improves downstream performance as well as retention of the original pretraining distribution, and that this operates in a domain and

---

[2]The different starting points are due to data availability, to ensure the midtraining mix does not repeat.

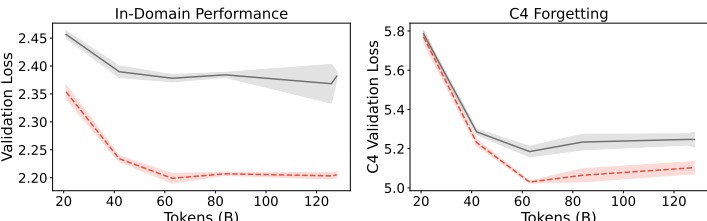

Figure 4: Validation loss and C4 loss for the Starcoder-midtrained model (160M) and base pretrained model *after* supervised fine-tuning on the Pycode dataset, with each point on the x-axis representing the number of tokens the pretrained checkpoint was trained on.

timing sensitive manner. However, this raises the question of how midtraining changes the model weights themselves. As a first step toward answering this question, we investigate changes after finetuning experienced by a midtrained as well as non-midtrained model in the code domain.

We use linear Centered Kernel Alignment (CKA) to measure layer-wise similarity between model states in the code domain (Kornblith et al., 2019). We extract activations from all layers using probe datasets (C4 and APPS (Hendrycks et al., 2021)) and compute CKA similarity matrices between four key model states: base pretrained, midtrained (Starcoder), base fine-tuned, and midtrained fine-tuned. If midtraining creates better representations for downstream tasks, we expect to see smaller representational changes during fine-tuning for midtrained models compared to base models.

Figure 5 shows the representational analysis for the 70M model. The midtrained model exhibits greater stability in the final layer after fine-tuning, a pattern consistent across model sizes (see Appendix G for the remaining results). However, the final fine-tuned models show high similarity regardless of whether models underwent midtraining. These effects are less pronounced for C4, which can be seen in Appendix H.

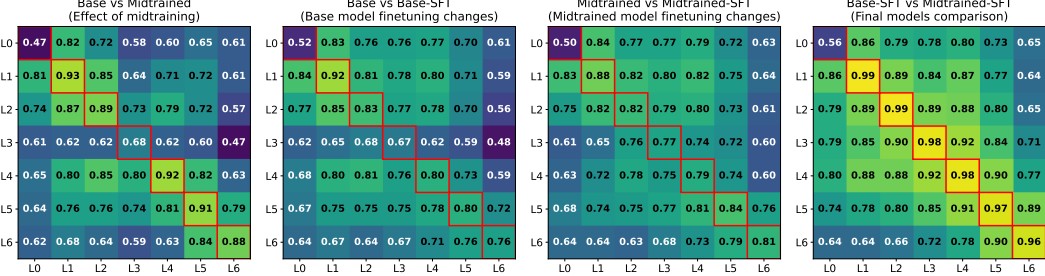

Figure 5: CKA analysis of model activations in the 70M model, probed with the APPS code dataset.

> **Finding 4.** Models exposed to midtraining require smaller representational shifts during fine-tuning, especially in the final layer, indicating smoother adaptation.

## 8 RELATED WORK

**Specific midtrained models** Recently, several language model families have adopted midtraining approaches with varying implementation details (Hu et al., 2024b; Dubey et al., 2024; OLMo Team et al., 2025; Chameleon Team, 2024). The midtraining phase duration varies from 2% (Hu et al., 2024b) to 20% (Chameleon Team, 2024) of total training, motivating our systematic investigation of timing effects. Common midtraining domains include code, math, instructions, and higher-quality web data (OLMo Team et al., 2025)—the domains we investigate. Beyond general-purpose models, midtraining has shown benefits for specific tasks like RL (Wang et al., 2025) and GUI agents (Zhang et al., 2025a). This widespread adoption motivates our questions of when and why midtraining provides downstream benefits.

**Multi-stage pretraining**   Several works explore multi-stage pretraining, Feng et al. (2024); Blakeney et al. (2024) focusing on two-stage pretraining and Zhang et al. (2025b) proposing four-stage pretraining. These approaches demonstrate improvements over single-stage pretraining. However, these works evaluate base model performance after pretraining, whereas we focus on the post-finetuning setting to focus on benefits that also affect posttraining.

**Continual Learning**   Domain-adaptive pretraining (DAPT) and related approaches continue pretraining on domain-specific data (Gururangan et al., 2020). Krishna et al. (2023) show that pretraining on downstream data alone can rival full pretraining when evaluated after fine-tuning, suggesting pretraining-posttraining alignment matters—consistent with our findings. Mehta et al. (2023) find pretraining reduces catastrophic forgetting during sequential fine-tuning; similarly, we observe midtrained models serve as better initializations with less forgetting.

**Stability in Continued Pretraining**   Recent work addresses stability challenges during continued pretraining. Guo et al. (2024) identify a "stability gap" where performance temporarily drops before recovering when shifting to new domains, Yang et al. (2024) synthesize larger training corpora from small domain-specific datasets, and Lin et al. (2024) introduce selective training on useful tokens only. While these works target training dynamics during continued pretraining, our approach examines how midtraining data selection affects post-fine-tuning performance, representing a complementary focus on end-task effectiveness.

**Relationship between Pretraining and Finetuning**   Several recent works have explored incorporating instruction-formatted data during pretraining. Allen-Zhu & Li (2023) show with an experiment on synthetic Wikipedia-style data that augmenting pretraining data with QA-formatted data improves subsequent fine-tuning, and Jiang et al. (2024) and Cheng et al. (2024) demonstrate this in a practical context as well. Sun & Dredze (2024) find continual pretraining benefits emerge only after fine-tuning, while Springer et al. (2025) show extended pretraining causes catastrophic forgetting ("overtraining"), particularly on math/code domains least aligned with web data. It is possible midtraining may prevent overtraining by introducing specialized data earlier and providing a better initialization for posttraining.

# 9   CONCLUSION

We conduct the first systematic investigation of midtraining through controlled experiments. We demonstrate that midtraining benefits are domain-specific, with the most substantial improvements in math and code domains that are not well represented in standard web pretraining corpora. Furthermore, we also find that midtraining mitigates catastrophic forgetting of general language modeling abilities after specific supervised fine-tuning and consistently outperformed continued pretraining on specialized data alone. Furthermore, timing of data introduction appears to have a stronger effect than mixture weight of that specialized data, though more work is needed to clarify this effect.

Based on these findings, we recommend that model trainers focus midtraining on domains which exhibit substantially different token patterns compared to the base pretraining data, given that they expect to posttrain on these domains. Furthermore, it may be worth it to begin the midtraining stage earlier than commonly practiced, in proportion to how different a domain is to general pretraining data. Lastly, even when adapting models to a specific domain, midtraining should take precedence over continued pretraining. However, despite these initial results, there is substantial future work to be done. A natural next step is to test the findings at much larger scales and with more diverse domains such as medicine, music, or other specific fields. Another direction is understanding how midtraining, and pretraining data distributions more broadly, interact with reinforcement learning based posttraining, and whether there are any differences between this and supervised finetuning based posttraining when it comes to adaptation. Finally, developing principled curricula that span multiple midtraining stages could shed light on how best to structure pretraining to prepare a model for posttraining while expanding its general knowledge. Further work in these directions could not only improve language model development but also improve our understanding of how data and curricula shape the emergence and stability of model capabilities.

## REPRODUCIBILITY STATEMENT

Anonymized data and code have been provided at the url `https://anonymous.4open.science/r/all_in_one_pretraining-0EF9`. Due to the size of pretraining data sources and checkpoints, these have not yet been uploaded to the cloud, but will be provided after acceptance. Additionally, multiple sections of the appendix describe pretraining and posttraining settings for ready reproduction (particularly Appendix A and Appendix B.

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

## A  PRETRAINING SETTINGS

We pretrained Pythia 70M, 160M, and 410M from scratch on the C4 dataset. All three models were trained for 128B tokens or approximately 61k steps, with very similar settings (documented in Table 4). L40S GPUs were used for all pretraining and midtraining runs. Models were trained with the LitGPT library (Lightning AI, 2023).

Table 4: Core pretraining hyperparameters for Pythia-70M, 160M, and 410M.

| Hyperparameter | 70M | 160M | 410M |
|---|---|---|---|
| Global batch size | 1024 | 1024 | 1024 |
| Micro batch size | 16 | 16 | 8 |
| LR schedule | Cosine w/ 10% warmup | Cosine w/ 10% warmup | Cosine w/ 10% warmup |
| Max LR | $3 \times 10^{-4}$ | $3 \times 10^{-4}$ | $3 \times 10^{-4}$ |
| Min LR | $1 \times 10^{-6}$ | $1 \times 10^{-6}$ | $1 \times 10^{-6}$ |
| Optimizer | AdamW | AdamW | AdamW |
| Betas | (0.9, 0.95) | (0.9, 0.95) | (0.9, 0.95) |
| Weight decay | 0.1 | 0.1 | 0.1 |
| Precision | BF16 | BF16 | BF16 |
| Num ranks | 4 | 4 | 8 |

## B  POSTTRAINING SETTINGS

We fine-tuned all models on four downstream datasets: Pycode (our 5K-sample subset of CodeSearchNet-Python), GSM8K (7.5K math problems), LIMA (1K instruction examples), and SciQ (13.7K science questions). For GSM8K only, the prompt/question portion was masked during loss; for the others the loss was computed over the full sequence. A summary of the datasets is given in Table 5. All runs used a cosine learning rate schedule with 10% linear warmup, trained for 4 epochs, global batch size 64, and micro-batch size 16 for 70M/160M (8 for 410M). Peak learning rates were selected by grid search on the base pretrained checkpoint before midtraining, and the LR grid is given in Table 6. Selected LRs for the final checkpoint of each model size are given in Table 7.

Table 5: Finetuning datasets.

| Dataset | # Train Samples | Prompt masked |
|---|---|---|
| Pycode (CodeSearchNet-Python subset) | 5,000 | No |
| GSM8K | 7,500 | Yes |
| LIMA | 1,000 | No |
| SciQ | 13,679 | No |

Table 6: Grid of candidate peak learning rates swept during tuning.

**LR grid**

```
4e-6, 8e-6, 1e-5, 2e-5, 4e-5, 5e-5, 6e-5, 7e-5, 8e-5, 9e-5, 1e-4, 1.2e-4,
1.4e-4, 1.6e-4, 1.8e-4, 2e-4, 2.4e-4, 4e-4, 5e-4, 6e-4, 8e-4, 1e-3, 2e-3,
3e-3, 4e-3, 6e-3
```

Table 7: Selected peak learning rates for fine-tuning (cosine schedule with 10% warmup).

| Dataset | 70M | 160M | 410M |
|---|---|---|---|
| GSM8K | 8e-4 | 4e-4 | 4e-4 |
| LIMA | 1.2e-4 | 5e-5 | 5e-5 |
| Pycode | 1e-3 | 5e-4 | 4e-4 |
| SciQ | 8e-4 | 2.4e-4 | 6e-4 |

## C  DATASET SIMILARITY MATRIX

We compute dataset similarity using surface-level token statistics after initial experimentation with embedding models gave implausible results for code datasets' similarities to other natural language datasets. For each pair of pretrain/midtrain and downstream datasets, we sample $\max(\text{dataset\_size}, 10{,}000)$ examples. Midtrain mixes are simulated by their actual compositions (e.g., Starcoder is treated as 20% Starcoder + 80% C4). From the (possibly mixed) texts we build unigram frequency vectors at a token level, normalize to probabilities, and compute: vocabulary Jaccard, overlap ratio, token-frequency cosine similarity, and a Jensen–Shannon-based similarity. These are combined as

$$\text{Combined} = 0.4 \cdot \text{cosine} + 0.3 \cdot \text{Jaccard} + 0.3 \cdot \text{JS\_similarity},$$

and used to fill the similarity matrix (diagonal entries are 1). This mixture-aware score reflects both specialty content and dilution by C4. Figure 6 shows the resulting similarity matrix between pre/midtrain datasets and SFT datasets.

## D  SFT IN-DOMAIN LOSS AND C4 LOSSES AFTER FINETUNING FOR 70M AND 160M MODELS

Table 8 depicts validation losses as well as C4 validation losses after finetuning on each SFT dataset, for the 70m and 160m models.

## E  MIDTRAINING LEG LENGTH VS. BENEFIT

Figure 7 shows the relationship between the two "legs" dist(C4, midtrain) and dist(midtrain, SFT) and the benefit of midtraining. As in Figure 2 this is computed based on the similarity matrix in Appendix C.

Table 8: SFT and C4 validation losses for 70m and 160m models across downstream datasets and midtraining mixtures, averaged across 5 seeds for each SFT dataset. Bold values indicate best performance within each dataset and model size combination.

| Model Size | Downstream Dataset | Midtrain Mix | SFT Val Loss | C4 Val Loss |
|---|---|---|---|---|
| 70m | Pycode | C4 | 2.633 | 6.067 |
| | | Starcoder (20%) | **2.530** | **5.996** |
| | | Math (12%) | 2.605 | 6.049 |
| | | FLAN (5%) | 2.648 | 6.112 |
| | | KnowledgeQA (20%) | 2.602 | 6.046 |
| | | DCLM (20%) | 2.592 | 6.007 |
| | GSM8k | C4 | 1.405 | 6.409 |
| | | Starcoder (20%) | 1.363 | **6.367** |
| | | Math (12%) | **1.359** | 6.391 |
| | | FLAN (5%) | 1.401 | 6.407 |
| | | KnowledgeQA (20%) | 1.373 | 6.434 |
| | | DCLM (20%) | 1.397 | 6.421 |
| | LIMA | C4 | 4.458 | 4.159 |
| | | Starcoder (20%) | 4.382 | 4.168 |
| | | Math (12%) | 4.418 | 4.180 |
| | | FLAN (5%) | 4.460 | 4.158 |
| | | KnowledgeQA (20%) | **4.371** | 4.152 |
| | | DCLM (20%) | 4.395 | **4.132** |
| | SciQ | C4 | 3.573 | 7.334 |
| | | Starcoder (20%) | 3.594 | 7.245 |
| | | Math (12%) | 3.599 | 7.289 |
| | | FLAN (5%) | 3.550 | **7.236** |
| | | KnowledgeQA (20%) | 3.537 | 7.268 |
| | | DCLM (20%) | **3.529** | 7.269 |
| 160m | Pycode | C4 | 2.382 | 5.247 |
| | | Starcoder (20%) | **2.205** | **5.104** |
| | | Math (12%) | 2.382 | 5.228 |
| | | FLAN (5%) | 2.389 | 5.253 |
| | | KnowledgeQA (20%) | 2.420 | 5.342 |
| | | DCLM (20%) | 2.405 | 5.268 |
| | GSM8k | C4 | 1.188 | 5.339 |
| | | Starcoder (20%) | 1.186 | 5.362 |
| | | Math (12%) | 1.139 | **5.198** |
| | | FLAN (5%) | **1.071** | 5.320 |
| | | KnowledgeQA (20%) | 1.190 | 5.346 |
| | | DCLM (20%) | 1.186 | 5.336 |
| | LIMA | C4 | **4.184** | 3.606 |
| | | Starcoder (20%) | 4.276 | **3.556** |
| | | Math (12%) | 4.421 | 3.599 |
| | | FLAN (5%) | 4.472 | 3.580 |
| | | KnowledgeQA (20%) | 4.498 | 3.566 |
| | | DCLM (20%) | 4.446 | 3.572 |
| | SciQ | C4 | 3.371 | 5.537 |
| | | Starcoder (20%) | 3.362 | 5.471 |
| | | Math (12%) | **3.316** | 5.437 |
| | | FLAN (5%) | 3.362 | 5.444 |
| | | KnowledgeQA (20%) | 3.339 | **5.373** |
| | | DCLM (20%) | 3.341 | 5.383 |

# F    REPRESENTATIVE TRAINING LOSS CURVES FOR MIDTRAINED VS. BASE MODELS

Figure 8 shows a representative training loss curve for a midtrained model when its domain is aligned to SFT data.

# G    ADDITIONAL CKA RESULTS ON APPS

Figure 9 and Figure 10 display the CKA layer similarity for 160m and 410m models.

# H    CKA RESULTS ON C4

Figure 11, Figure 12, and Figure 13 show the CKA layer similarity for all model sizes with C4 as a probe.

# I    STATEMENT ON LLM USAGE

Large language models (LLMs) were used to assist with refining writing in this submission, including summarizing paragraphs in order to shorten the submission, correcting grammar, and giving suggestions to improve organization. LLMs were not used in the ideation process and analyses and experimental setups were designed fully by the authors. Copilot and other coding agents were used to generate some utility scripts in the process of coding.

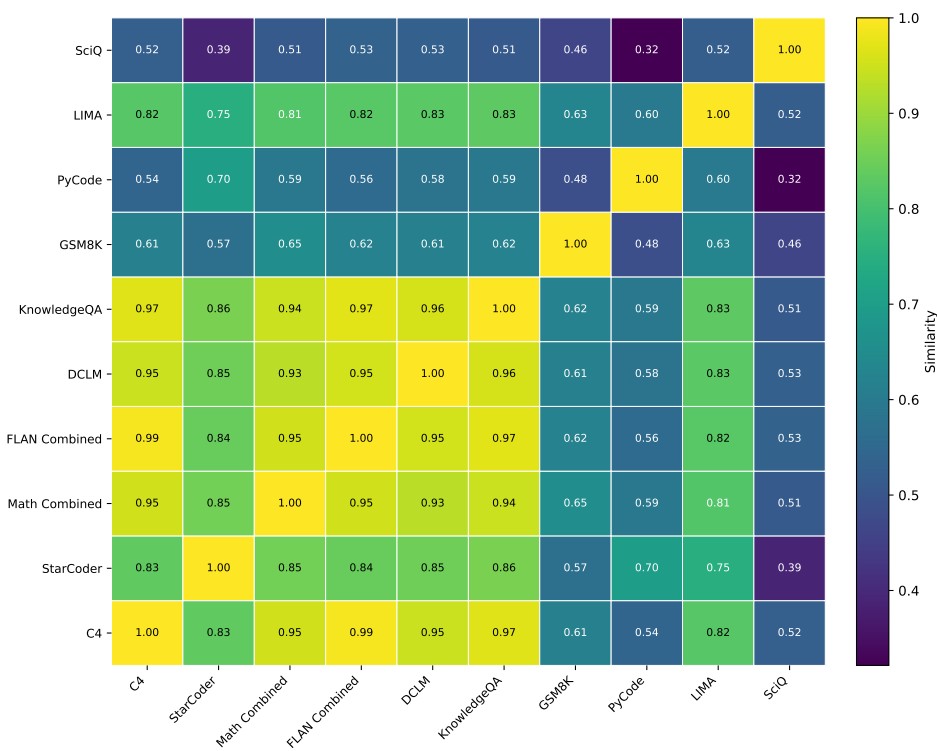

Figure 6: Token-based similarity matrix for pre/midtrain and SFT datasets. Note that these midtrain datasets are corrected for mix weight in this matrix.

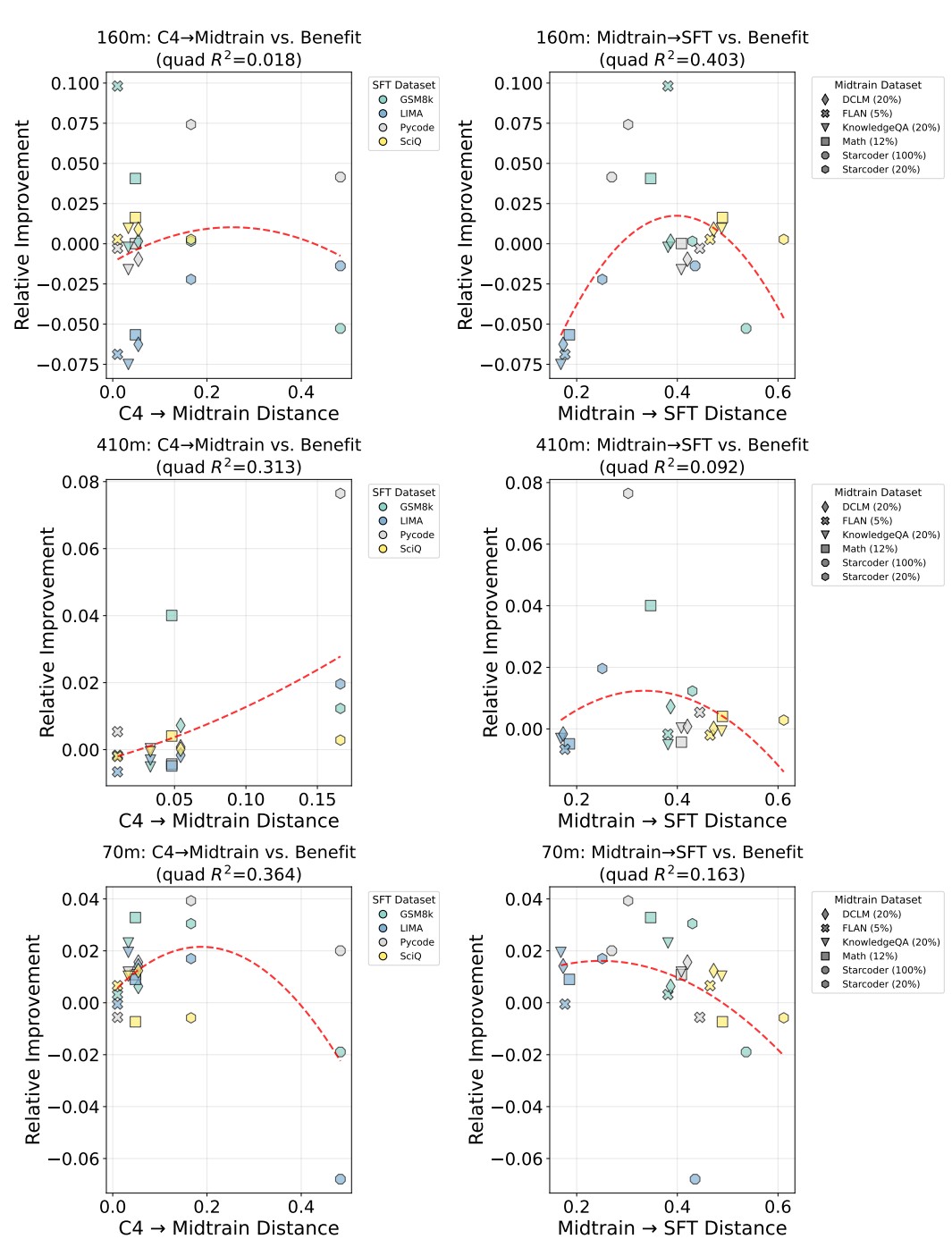

Figure 7: Relationship between the two "legs" dist(C4, midtrain) and dist(midtrain, SFT) and the benefit of midtraining.

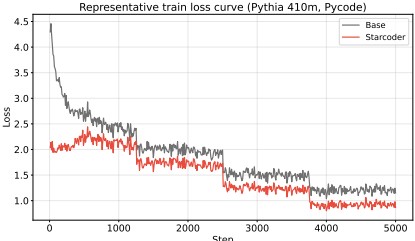

Figure 8: Representative training loss curve for a midtrained model and base model on Pycode, for Pythia-410m. The midtrained model starts with a lower training loss, and maintains a slight gap throughout training.

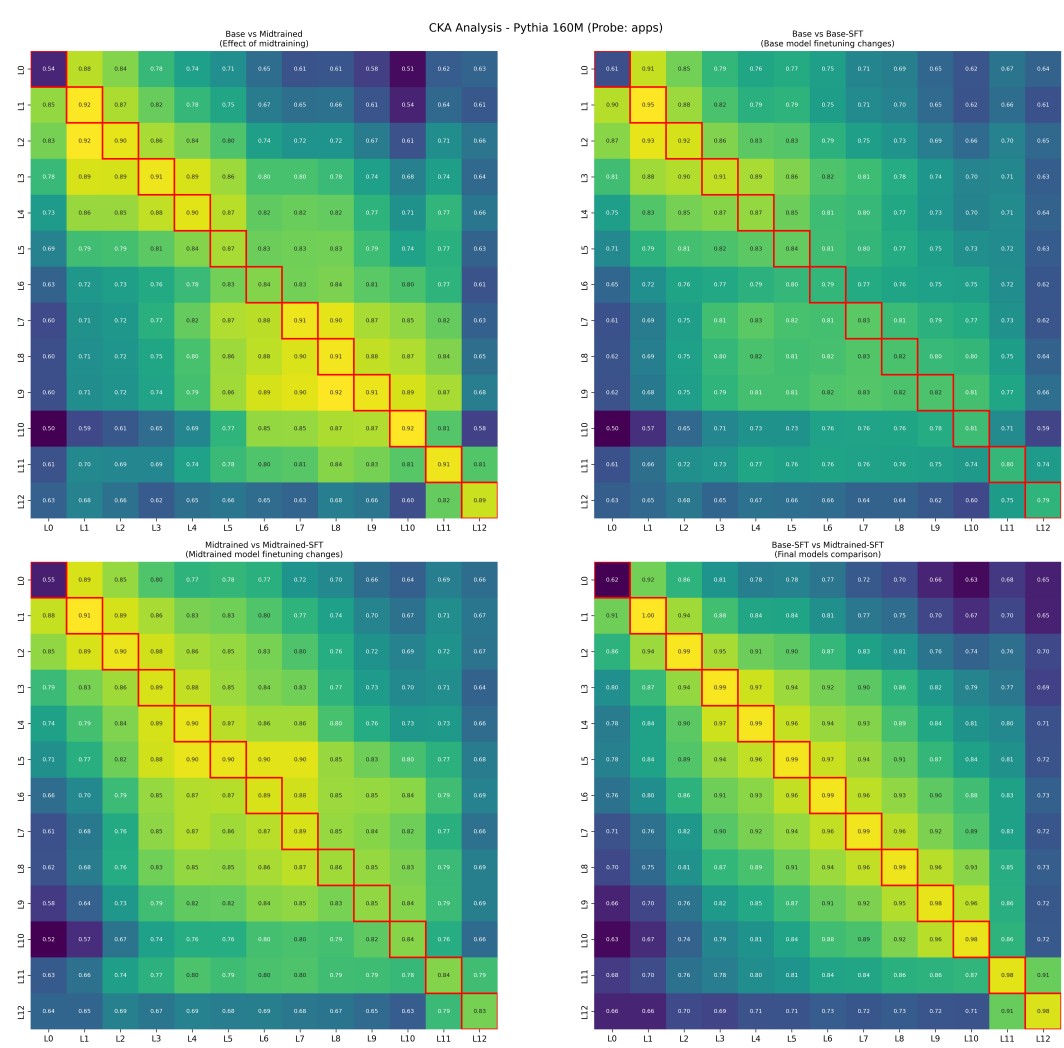

Figure 9: CKA layer analysis for Pythia-160M with APPS as a probe.

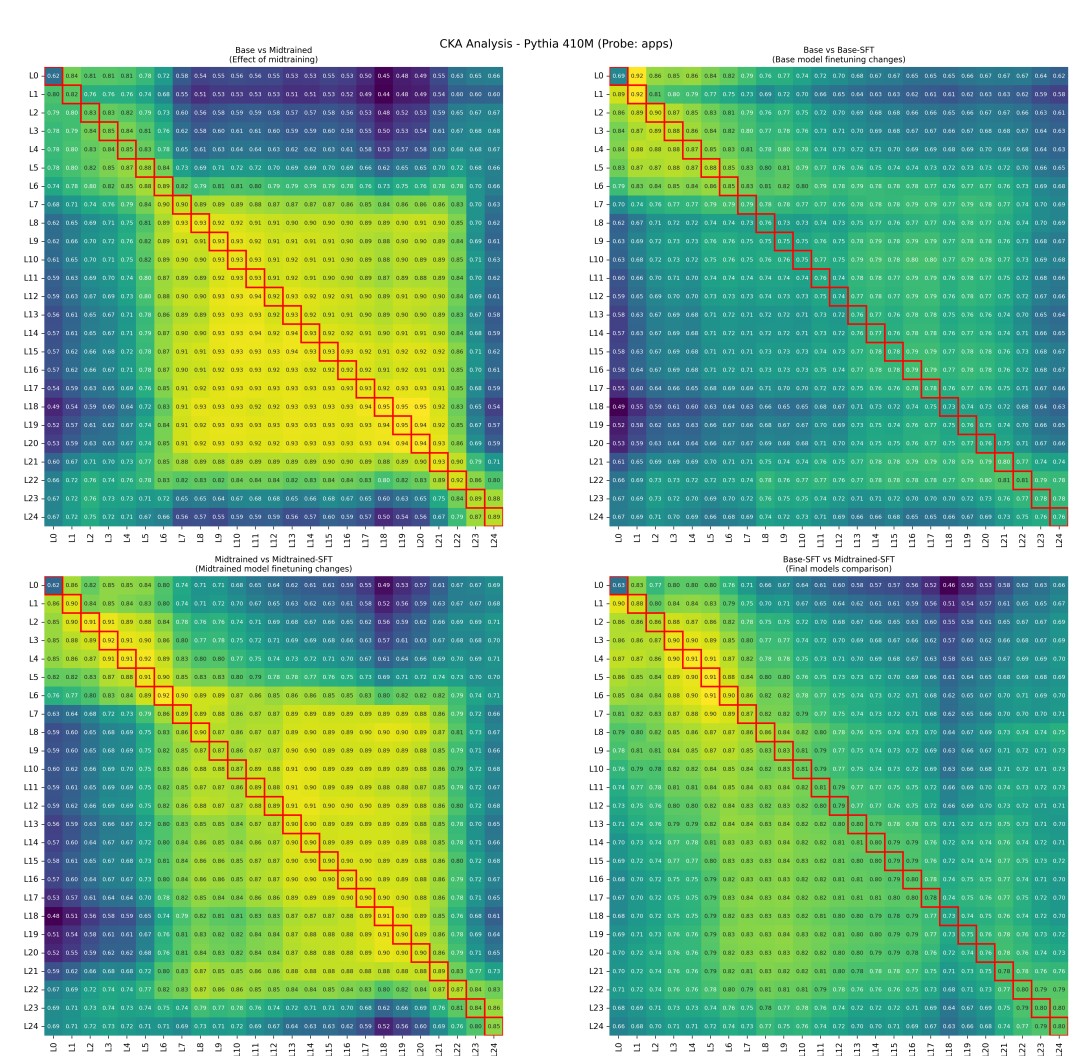

Figure 10: CKA layer analysis for Pythia-410M with APPS as a probe.

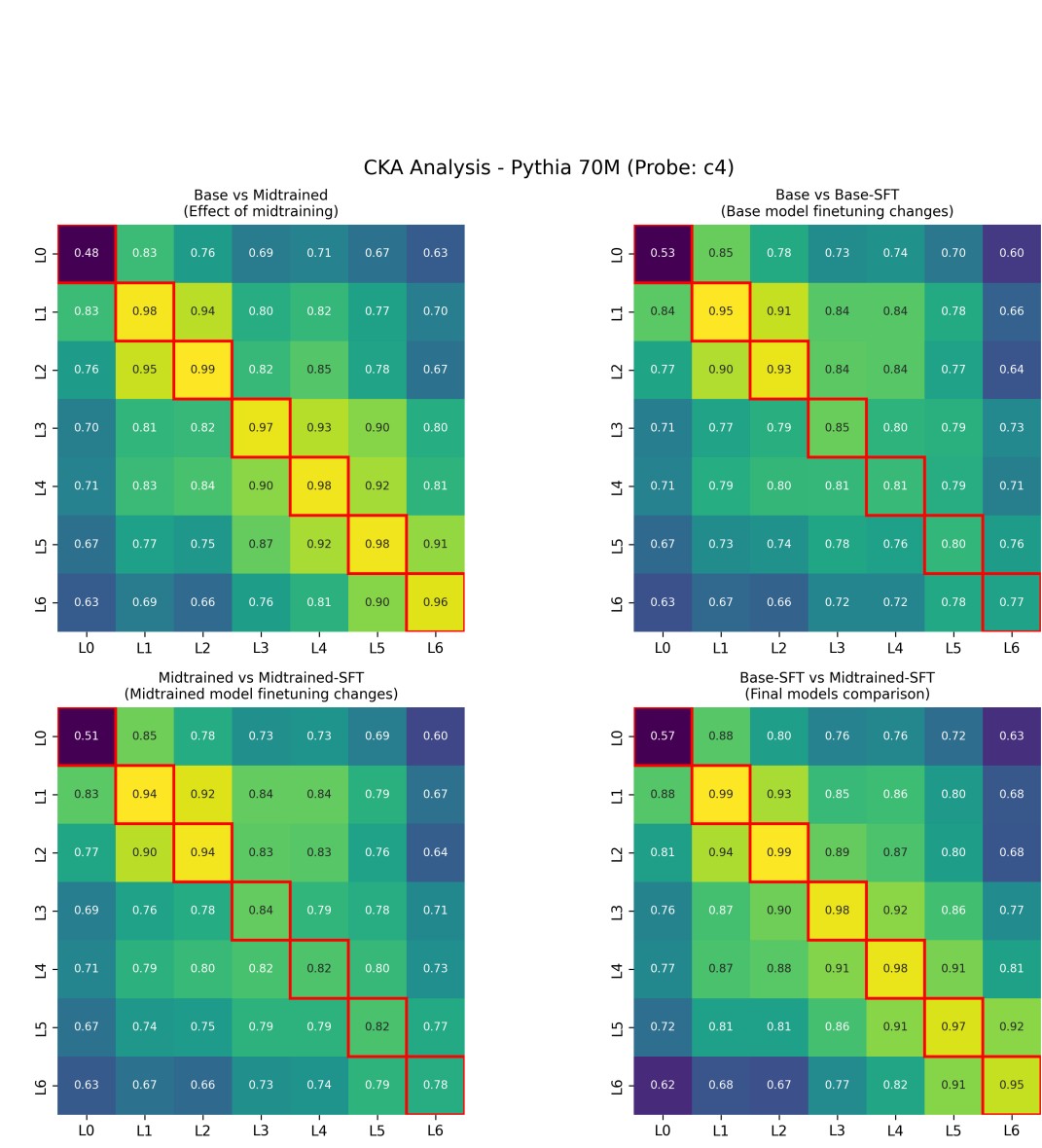

Figure 11: CKA layer analysis for Pythia-70M with C4 as a probe.

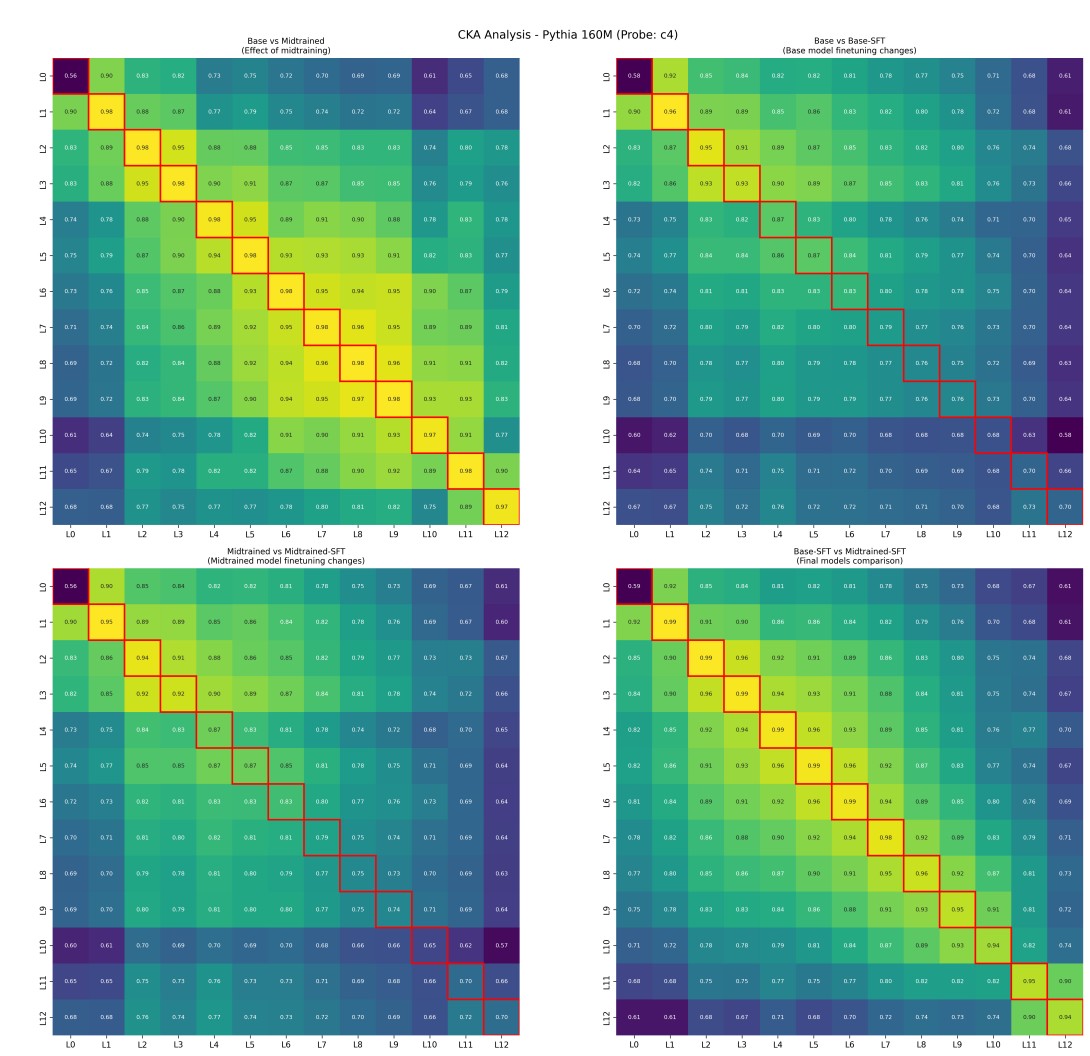

Figure 12: CKA layer analysis for Pythia-160M with C4 as a probe.

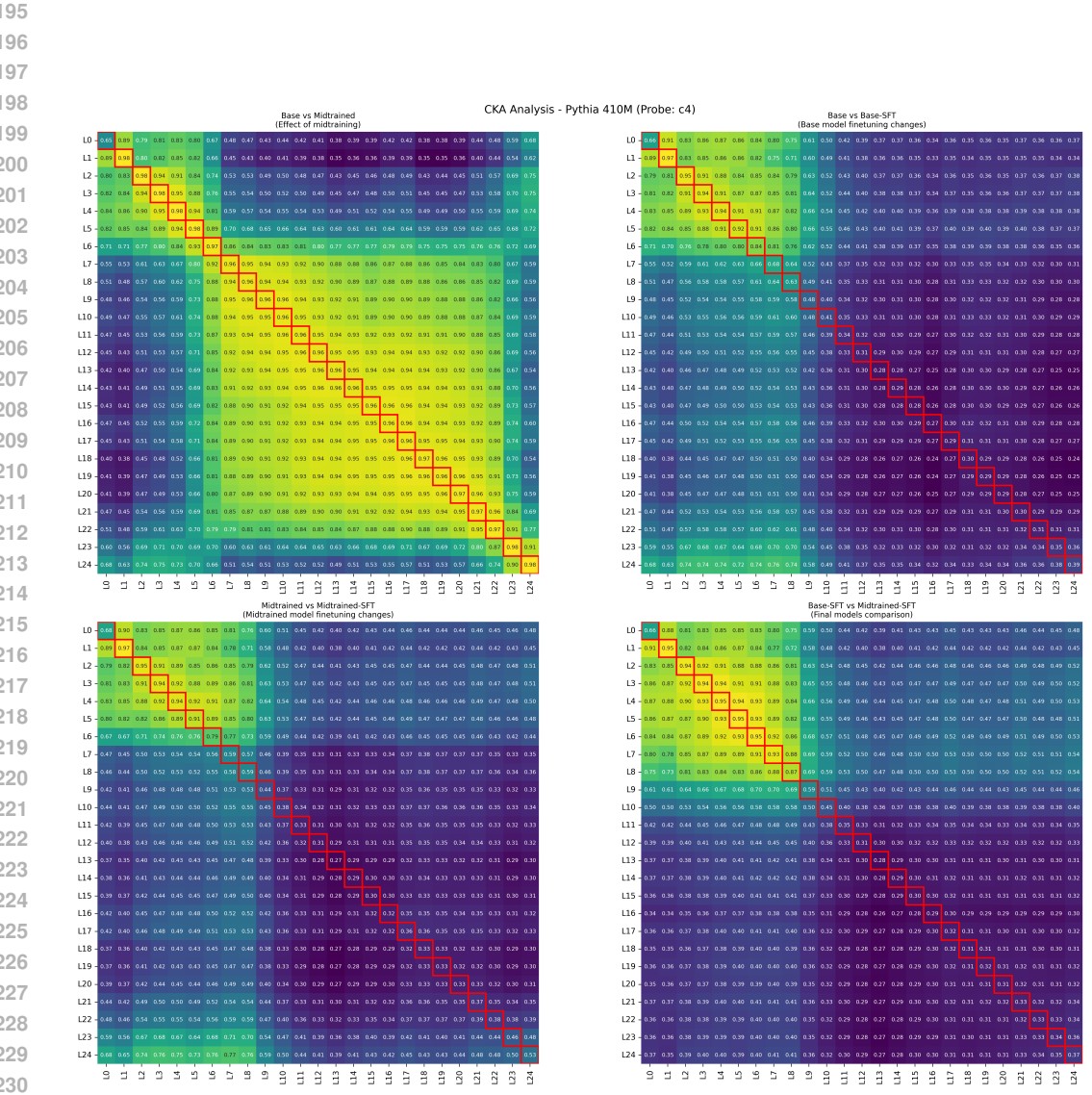

Figure 13: CKA layer analysis for Pythia-410M with C4 as a probe.

