# OpenReview forum: "Midtraining Bridges Pretraining and Posttraining Distributions"
_ICLR.cc/2026/Conference — Submitted to ICLR 2026_

### Official Review · Reviewer_FGaf · 2025-10-31

**Soundness:** 2
**Presentation:** 2
**Contribution:** 2
**Rating:** 2
**Confidence:** 5

**Summary:**

This paper measures the effectiveness of changing the pretraining distribution mid-training. The problem is formalized as picking a starting point for mid-training and a mixing parameter between the mid-training distribution and the initial pretraining distribution. The experiment explores these settings for 4 test domains and 3 model sizes.

**Strengths:**

* Clear presentation of the motivations, an important topic for LLM training.
* Experiments are performed over a variety of domains.

**Weaknesses:**

* The paper does not report a joint hyper-parameter search over the two hyper parameters of midtraining (fraction of pretraining data in the second phase, fraction of total tokens in the second phase). The reader cannot understand how the two parameters interact, e.g. could a shorter 2nd phase might be compensated by seeing less pretraining data in the second phase? It is not even clear if any pretraining is necessary, i.e. could the best result be obtained with a single training phase that mixes in-domain and pretraining data (single mixture).
* The paper ignores prior work on injecting pretraining data during supervised fine tuning which is a lot like your definition of mid-training, including
   * Liu et al. Improved fine-tuning by better leveraging pretraining data. Advances in Neural Information Processing Systems, 35:32568–32581, 2022.
   * Kang et al Get more for less: Principled data selection for warming up fine-tuning in llms. arXiv preprint arXiv:2405.02774, 2024.
   * Ibrahim et al Simple and scalable strategies to continually pre-train large language models, 2024.
   * Bethune et al. Scaling laws for forgetting during fine-tuning with pretraining data injection. arXiv preprint arXiv:2502.06042, 2025.
* The paper provides only limited scale experiments and does not attempt at extracting a scaling law to predict behaviour at scales larger than their computing budget.

**Questions:**

* L018 “Syntactic gap” what do you mean? Could you define it?
* L133 “models have stabilized" what is a stabilized model? Could you define it?
* The legend is missing on Figure 4. Could you include it?
* The font in Table 1 is smaller than the main text font.
* Figure 3 is too small, esp. the figures are not tall enough to be readable. Most of the labels (titles, axis names and legends) are repeated several times: space can be used more efficiently.
* In Table 3. How were the fraction of starcoder/math selected? How was the amount of pretraining vs mid-training selected? How do these two parameters interact?
* In Figure 3b and 3c, it seems that the 20% mix of starcoder is not particularly good for both model sizes. Why did you pick 20% for the other experiments?
* In Table 2 and Table 8: it seems that the best data source for LIMA is different for all three model sizes, how do you explain this inconsistency?

---

> ### Author Response · Authors · 2025-12-03
> **Response to weaknesses**
>
> We thank the reviewer for recognizing the importance of our topic and the variety of domains we explore. However, we respectfully disagree with several characterizations in the review and believe some requests misunderstand our contribution.
>
> > The paper does not report a joint hyper-parameter search over the two hyper parameters of midtraining (fraction of pretraining data in the second phase, fraction of total tokens in the second phase). The reader cannot understand how the two parameters interact, e.g. could a shorter 2nd phase might be compensated by seeing less pretraining data in the second phase? It is not even clear if any pretraining is necessary, i.e. could the best result be obtained with a single training phase that mixes in-domain and pretraining data (single mixture).
>
> We want to clarify the experiments we have conducted. We systematically explored both timing and mixture weight for the code domain (Figure 3):
>
> Timing ablation (Fig 3a): 20% mixture, 5 different start points
> 12B, 42B, 63B, 84B, 105B
> Mixture ablation at 105B (Fig 3b): 5 different mixtures
> 10%, 20%, 30%, 50%, 80%
> Mixture ablation at 63B (Fig 3c): 4 different mixtures
> 20%, 30%, 50%, 80%
> These ablations span 12 different experimental conditions. We have made efforts to show how timing and mixture weight interact. We agree that it would be ideal to do a full grid search over timing and mixture weight, but this would require full pretraining of dozens of models on top of our current experiments.
> We will add the following experiments for ICML to address these concerns: First, we'll train models from scratch with a constant 20% code mixture throughout all of pretraining (no separate midtraining phase). This directly tests whether the phased approach is necessary or if a single mixed phase works just as well. Second, we'll run additional timing and mixture weight combinations to better understand how these factors interact. In particular, we want to test whether high mixture weights at later timepoints can compensate for delayed introduction of specialized data. These experiments will help us definitively answer whether timing is fundamental to midtraining or whether it can be traded off against mixture weight.
>
> > The paper ignores prior work on injecting pretraining data during supervised fine tuning which is a lot like your definition of mid-training, including [...]
>
> We thank you for finding these references and will add them to our related work. However, we disagree that these works address the same problem. In short, the Liu et al, Kang et al and Bethune et al inject pretraining data during supervised fine-tuning rather than at the end of pretraining, while the Ibrahim et al paper studies continual pretraining. Our work focuses on strategic data mixing during the initial pretraining stage to prepare for downstream fine-tuning. These are complimentary approaches as one could potentially do both midtraining as well as replay of pretraining data during finetuning.
>
>
> > The paper provides only limited scale experiments and does not attempt at extracting a scaling law to predict behaviour at scales larger than their computing budget.
>
> We agree that it is important to test findings at multiple scales. However, the focus of our work is not immediately on constructing a scaling law. One core finding that we highlight is that midtraining scaling laws need to capture the three-way interaction between pretraining data, midtraining configuration, and posttraining data, where effects can best be observed after fine-tuning on a specific dataset. Essentially this would involve extending standard scaling laws to capture curriculum effects. This is an important open problem that our work makes tractable by establishing: (1) which factors matter (timing > mixture weight, distributional distance), (2) how to measure relevant properties (our similarity metric).
> Since midtraining effectiveness depends on all three factors, constructing midtraining scaling laws would represent a considerable computational undertaking that we will leave for future work. We emphasize that this is the first paper formally defining midtraining and conducting initial controlled experiments, which provides a foundation for future work and followups such as extracting a scaling law.

---

### Official Review · Reviewer_x5YP · 2025-11-01

**Soundness:** 2
**Presentation:** 3
**Contribution:** 3
**Rating:** 4
**Confidence:** 2

**Summary:**

This paper systematically examines midtraining, an intermediate phase that mixes specialized data into pretraining to improve posttraining outcomes. Through controlled experiments , the authors test various midtraining domains (code, math, instructions, QA) and analyze effects on downstream performance and forgetting. They find that midtraining is most effective in domains far from web-text data, acting as a bridge between pretraining and fine-tuning distributions. Early introduction and mixed data outperform late or domain-only approaches. The study provides an empirical foundation for understanding midtraining as a structured domain adaptation technique.

**Strengths:**

- The paper provides a clear and well-executed study of midtraining, a timely and underexplored topic in large language model development. Its originality lies in empirically evaluating this training phase across domains.

- The experiments are thorough, well controlled, and convincingly analyzed.

- The writing is clear and structured, with results that directly support the claims.

- The work offers meaningful insights into how midtraining bridges pretraining and fine-tuning, making a valuable and practical contribution to understanding domain adaptation in modern model training.

**Weaknesses:**

- While the study is thorough, its scope is limited to relatively small models (up to 410M parameters), leaving uncertainty about how the findings scale to contemporary billion-parameter systems. The paper would benefit from stronger evidence that the observed trend particularly regarding timing sensitivity and mixture effects hold at larger scales.

- The analysis focuses on supervised fine-tuning and does not test whether the same principles apply to reinforcement learning or preference-based posttraining, which are now standard in practice. Including even limited experiments or discussion in this direction would strengthen the generality of the conclusions.

- The paper would benefit from citing and briefly discussing related work that also explores interventions during pretraining to improve posttraining adaptability. In particular,  “Improving Language Plasticity via Pretraining with Active Forgetting” (NeurIPS 2023) proposes a complementary approach that modifies pretraining dynamics to enhance model plasticity. While the mechanism differs from midtraining, both works share the motivation of bridging or improving transitions between pretraining and fine-tuning phases. Adding a short comparison (e.g., in the Related Work section) would help position this paper more clearly within the broader literature on training-phase modifications for improved adaptation.

**Questions:**

1. How might the effects of midtraining, particularly the importance of timing, extend to larger models?
2. Would the same bridging behavior be expected if posttraining were done through reinforcement learning or preference optimization rather than supervised fine-tuning?
3. How do the authors view midtraining in relation to other ways of shaping pretraining dynamics, such as active forgetting or data reweighting? A brief reflection on their connections would help clarify its conceptual role.

---

> ### Author Response · Authors · 2025-12-03
> **Response to weaknesses**
>
> We thank you for your appreciation of our work as clean and well-executed, and for your appreciation of our experiments as thorough, well controlled, and convincingly analyzed! We address your concerns below.
>
> > While the study is thorough, its scope is limited to relatively small models (up to 410M parameters), leaving uncertainty about how the findings scale to contemporary billion-parameter systems. The paper would benefit from stronger evidence that the observed trend particularly regarding timing sensitivity and mixture effects hold at larger scales.
>
> We agree that it would make sense to include results from larger models to see if findings hold at scale.  We are currently in the process of training and evaluating a 1B parameter model across our key experimental conditions. The pretraining runs are complete, and we are conducting fine-tuning and evaluation on our benchmark suite. Complete results will be included in the ICML submission and will test whether our findings about timing effects and mixture weights hold at scales closer to contemporary practice, providing important evidence about the stability of our patterns across the 160M-1B range.
>
> However, we want to strongly emphasize the computational requirements of our experimental design. Especially for the timing and mixture weight ablations, each experimental condition requires full pretraining from scratch, followed by fine-tuning and evaluation across multiple seeds. each 1B model required approximately 1,000-1,200 GPU-hours per condition (5-6 days on 8× L40S GPUs) for pretraining alone, plus additional time for fine-tuning and evaluation across multiple seeds.
>
>
> > The analysis focuses on supervised fine-tuning and does not test whether the same principles apply to reinforcement learning or preference-based posttraining, which are now standard in practice. Including even limited experiments or discussion in this direction would strengthen the generality of the conclusions.
>
> Thank you for this suggestion, we agree that it would be interesting to see if the same conclusions hold with RL based posttraining. However, we respectfully consider RL and preference based posttraining to be out of scope for this work. Our focus was to establish an initial systematic study of midtraining, and we will consider extending this to RL based posttraining in future work.
>
> On the generalizability front, we believe our findings should extend to RL-based posttraining because the core mechanism of bridging distributional gaps between pretraining and posttraining data (Figure 2, Section 5) and reducing representational shifts during adaptation (Figure 5, Section 7) operates at a level that should be agnostic to the specific posttraining objective. Evidence for this can be seen in recent work by Wang et al. (2025) "Octothinker: Mid-training incentivizes reinforcement learning scaling" which demonstrates midtraining has benefits after RL posttraining as well.
>
> We choose to focus on supervised fine-tuning for this paper because it was the initial context in which midtraining benefits were reported (as the Wang 2025 paper was the first one to study midtraining in an RL context as far as I am aware). We will add discussion of this limitation to the paper, but we believe that this would be a valuable follow-up study rather than a strict requirement of establishing midtraining effects.
>
> > The paper would benefit from citing and briefly discussing related work that also explores interventions during pretraining to improve posttraining adaptability. In particular, “Improving Language Plasticity via Pretraining with Active Forgetting” (NeurIPS 2023) proposes a complementary approach that modifies pretraining dynamics to enhance model plasticity. While the mechanism differs from midtraining, both works share the motivation of bridging or improving transitions between pretraining and fine-tuning phases. Adding a short comparison (e.g., in the Related Work section) would help position this paper more clearly within the broader literature on training-phase modifications for improved adaptation.
>
>
> Thank you for bringing this paper to our attention, we will add it to our related work and discussion.

---

### Official Review · Reviewer_To7W · 2025-11-07

**Soundness:** 2
**Presentation:** 3
**Contribution:** 1
**Rating:** 2
**Confidence:** 3

**Summary:**

The paper study the impact of midtraining on the final performance, in particular through the lens of the similarity between pretraining set and midtraining set, and between midtraining and posttraining (SFT). Realistic high quality datasets are used, such as Staroder or DCLM. The model range remains 410M, using the Pythia family.

The performance is evaluated on some specialized domains using the validation loss

**Strengths:**

### Timely problem

The problem by the paper is timely, as midtraining is becoming increasingly important in learning pipelines.

### Realistic datasets

These datasets have been used in the past for pretraining or midtraining of sucessful models.

### Experiments of Sec. 6

I think the experimental setup of Sec. 6 is very promising, as the dilemma "mixture VS time" is paramount to these considerations.

### CKA

The tools of Section 7 (CKA) are also interesting, and might provide good explanations.

**Weaknesses:**

### Lack of accuracy-based benchmarks

The model scale limits the possibility of relying on downstream providing an accuracy, since models below 410M typically operate at the random level on many tasks. However, a non-random accuracy can be expected on tasks like ARC-Easy, so it could be interesting to measure performance through that lens.

This weakens impact of the paper, as improvement/alignment as maily measured through distributional similarity at the token logits level. This measure is based on adhoc cosine + Jaccard + Jensen-Shannon combination, but it is unclear why this metric makes sense. KL divergence looks like a more sensical choice.

### Unsurprising conclusions

The similarity measure is a bit adhoc, and some conclusions (like finding 1) are expected.

> Finding 3. Maintaining a mixture with general data in midtraining is preferable to continued
pretraining on specialized data alone.

This is also a finding documented in:

"Ibrahim, A., Thérien, B., Gupta, K., Richter, M.L., Anthony, Q., Lesort, T., Belilovsky, E. and Rish, I., 2024. Simple and scalable strategies to continually pre-train large language models. arXiv preprint arXiv:2403.08763."

> Finding 4. The timing of midtraining is more critical than the mixture weight; early intro-
duction of specialized data in the code domain leads to stronger in-domain gains and better
retention of general capabilities.

I think we miss more ablations to conclude this: other midtraining/pretraining/postraining mixtures could have yield a different conclusion. See "questions".

**Questions:**

**Q1:** What is the rational between the similarity measure?

**Q2:** do you have a downstream benchmark accuracy (like ARC easy) results for some of these models?

**Q3:** in section 6. can you report results as total number of tokens seen from each domain, instead of training time? At equal number of midtraining tokens, which setup wins between early and late midtraining (playing on relative ratios to achieve this)?

**Q4:** in Sec 7., is there a correlation between CKA and the metric you define in Appendix C?

---

> ### Author Response · Authors · 2025-12-03
> **Response to weaknesses (1/2)**
>
> Thank you for appreciating the timeliness of our work, the use of realistic datasets, and the experimental setup in comparing timing of data introduction and mixture weight of specialized data. We address each concern below:
>
> > A. The model scale limits the possibility of relying on downstream providing an accuracy, since models below 410M typically operate at the random level on many tasks. However, a non-random accuracy can be expected on tasks like ARC-Easy, so it could be interesting to measure performance through that lens.
>
>
> We agree that this would be a valuable addition. We have no evaluated our models on ARC-Easy as suggested:
>
> Our 410M models achieve 46.51% accuracy on ARC-Easy for baseline pretraining and 47.73% for code midtraining. This 1.2 percentage point improvement from midtraining, while modest, demonstrates measurable benefits even at this scale. We will include complete results across all our model sizes in the revised manuscript.
>
> > B. This measure is based on adhoc cosine + Jaccard + Jensen-Shannon combination, but it is unclear why this metric makes sense. KL divergence looks like a more sensical choice.
>
> We appreciate this feedback and also acknowledge that our metric may initially appear ad-hoc. However, we want to clarify our design choices and alternatives that we explored.
>
> Regarding KL divergence, we appreciate the suggestion but respectfully note that we are already using JS divergence, which is closely related to KL divergence. Specifically,
>
> $\text{KL}(P||Q) = \sum_i p_i \log(p_i/q_i)$
> $\text{JS}(P,Q) = \frac{1}{2}\text{KL}(P||M) + \frac{1}{2}\text{KL}(Q||M)$, where $M = \frac{1}{2}(P+Q)$
>
> We chose JS divergence because of its properties (symmetry, boundedness, and numerical stability when probabilities are near 0 vs. KL divergence).
>
> We additionally experimented with several alternatives before settling on our combined metric:
>
> - **Embedding based similarity** (using Qwen-3-8B): we tried several models from the MTEB leaderboard, with Qwen3-8B shown. However, these produced implausible results, mostly due to not representing code domains well. For instance, one sanity check is that we would expect Starcoder data to be similar to CodeSearchnet-python, moreso than to other datasets, but this was not the case. This may have been because the embeddings may not have been trained well to distinguish code from natural language.
>
> Example of embedding based similarity: [Image linked here](https://imgur.com/a/VzdgB6q)
>
> - **Gradient based similarity** (comparing gradients from the base model with respect to samples from the different datasets): this was highly unstable across different models, layers, and samples from datasets. This instability made it unreliable as a consistent similarity measure.
>
> Example of gradient based similarity: [Image linked here](https://imgur.com/a/gdqnOUk)
>
>
> Our combined metric was designed to be simple and capture three complimentary aspects of dataset similarity:
>
> Jaccard (30%): vocabulary overlap, which clearly separates datasets that use very different vocabulary items (e.g. natural language and code)
> Cosine (40%) + JS divergence (30%): overall distributional similarity of token frequencies, from slightly different perspectives
>
> Lastly, we emphasize that the key validation for this metric is empirical: it successfully predicts midtraining effectiveness (figure 2, r=0.869, p<0.001 for 70M etc). While more complex measures may be possible, this suggests that our measure already captures a sufficient notion of distributional similarity for practical purposes. If you have additional alternative measures that would be more principled, we would be happy to evaluate them and add these to the paper as well.

---

> ### Author Response · Authors · 2025-12-03
> **Response to weaknesses (2/2)**
>
> > Finding 3. Maintaining a mixture with general data in midtraining is preferable to continued pretraining on specialized data alone.
> This is also a finding documented in:
> "Ibrahim, A., Thérien, B., Gupta, K., Richter, M.L., Anthony, Q., Lesort, T., Belilovsky, E. and Rish, I., 2024. Simple and scalable strategies to continually pre-train large language models. arXiv preprint arXiv:2403.08763."
>
> While we appreciate this reference and will add it to our related work section, we would like to respectfully point out that this paper addresses a different setting with distinct findings. This paper studies continual pretraining of an already pretrained model when new data becomes available, and their key finding is that replaying the original training data alongside new data prevents catastrophic forgetting.
>
> Crucially, they evaluate performance directly on the pretrained models by measuring validation loss and benchmark performance without any downstream fine-tuning.
> Our work studies midtraining during initial training from scratch and evaluates performance after supervised fine-tuning. Our Finding 3 (Table 3) compares three approaches measured post-SFT: (1) no midtraining (baseline pretraining), (2) mixed midtraining (ongoing 20% domain + 80% general mixture), and (3) continued pretraining (switch to 100% domain data). We find that mixed midtraining outperforms both alternatives on in-domain SFT performance while better retaining C4 capabilities post-SFT.
>
> The experimental settings differ fundamentally: Ibrahim et al. start with a pretrained checkpoint and adapt it to new data, measuring base model performance. We train from scratch with strategic data ordering and measure how well models adapt to downstream SFT tasks. While both works involve data mixing, ours addresses the question of how we should structure pretraining data to optimize for subsequent fine-tuning rather than how we should update pretrained models with new data.
>
> > Finding 4. The timing of midtraining is more critical than the mixture weight; early intro- duction of specialized data in the code domain leads to stronger in-domain gains and better retention of general capabilities.
> I think we miss more ablations to conclude this: other midtraining/pretraining/postraining mixtures could have yield a different conclusion. See "questions".
>
> We appreciate this concern about the completeness of our ablations. We want to clarify what experiments we have conducted while also acknowledging the limitations of not running a full grid search.
> The experiments in the current paper cover the following settings:
>
> Timing ablation (Fig 3a): 20% mixture, 5 different start points
> 12B, 42B, 63B, 84B, 105B
> Mixture ablation at 105B (Fig 3b): 5 different mixtures
> 10%, 20%, 30%, 50%, 80%
> Mixture ablation at 63B (Fig 3c): 4 different mixtures
> 20%, 30%, 50%, 80%
>
> These ablations span 12 different experimental conditions, representing a substantial computational investment given that each condition requires full pretraining from scratch plus finetuning.
> What we observe is that the timing effect (Fig 3a) shows much larger performance differencescompared to mixture weight effects at either 105B (Fig 3b) or 63B (Fig 3c). Additionally, Figure 4 shows that in-domain benefits emerge quickly after midtraining introduction while C4 retention benefits develop more gradually, further supporting the importance of timing.
>
> We agree that a complete grid search over all timing and mixture weight combinations would be ideal. However, this would require hundreds of additional pretraining runs, which is computationally prohibitive at our scale. Our current ablations represent strategic "slices" through this parameter space that we believe capture the main effects. We also acknowledge that our findings are specific to the code domain and other domains may exhibit different sensitivities.
>
> For the updated version, we will conduct a complete grid search over timing and mixture weight combinations to address this concern.
>
> Finally, we will revise our claim to be more precise about the scope (code midtraining mix, and specific settings we tried).
>
> The broader point of Finding 4 is that practitioners should carefully consider when to introduce specialized data during pretraining, not just how much to include, which has not received as much attention in past work. While we cannot claim this holds universally across all mixes or configurations, our systematic ablations across multiple timing points and mixture weights provide evidence for this principle in the code domain.

---

> ### Author Response · Authors · 2025-12-03
> **Response to questions**
>
> Q1: See response to weaknesses 1b.
> Q2: See response to weaknesses 1a.
>
> Q3:This is a great question. Since C4 contains no code, all code exposure comes purely from the midtraining phase, so total code tokens are already controlled.
>
> Importantly, even with the existing results, we already have evidence that timing matters beyond just total exposure. In Figure 3b, we tested mixture weights from 10% to 80% at the 105B starting point. Even with 80% code mixture starting at 105B (which gives substantial total code exposure), performance is still worse than starting earlier with only 20% mixture. For the 160M model: - 12.6B start with 20% mixture: validation loss 2.205, ~23B code tokens total - 105B start with 80% mixture: validation loss 2.300, ~18B code tokens total - 105B start with 20% mixture: validation loss 2.333, ~5B code tokens total So even though the 105B/80% condition sees nearly as much code as the 12.6B/20% condition (18B vs 23B code tokens), it still performs substantially worse (2.300 vs 2.205). This suggests that timing has an independent effect beyond total domain exposure. We will make this point more explicit in the revised manuscript and add clear reporting of total code tokens for each condition to make this analysis
>
> Q4: This is an interesting question that we have not yet investigated. We will do this analysis in future work.

---

### Meta-Review · Area_Chair_e1TX · 2026-01-02

**Summary:**

The author inspecting the impact of midtraining from various perspectives: the starting point of mid-training, the similarity between pretraining and midtraining, and midtraining and posttraining (SFT). Experiments conducted on various domains (code, math, instructions, QA) with model sizes up to 410M. Authors find that midtraining is most effective in domains far from web-text data, acting as a bridge between pretraining and fine-tuning distributions; early introduction and mixed data outperform late or domain-only approaches. The study provides an empirical foundation for understanding midtraining as a structured domain adaptation technique

Reviewers found the paper is of following strength
* Study the timely problem to understand training dynamics behind LLMs
* Proper selection of training datasets
* Experiments are performed over a variety of domains, and support the claims/conclusion made in the paper to offer insights into how midtraining bridge pretraining and posttraining
* The paper is very clear and well-written.

**Reviewer Concerns:**

We also found the additional explanation from authors during rebuttal are helpful (e.g., discussing relationship between this work and others, explaining the selection and design of experiments and the scope of this paper, explaining more methodology details and insights). These are very helpful discussion and will strengthen this work.

However, the reviewers found impact of this work is below the bar for a ICLR paper, because of the limited scale of experiments. Compared to mainstream LLMs, 410M model size is too small to extract meaningful conclusions about training dynamics and performance. Many capabilities only emerge after scaling models to billions of parameters. Reinforcement learning, as popular posttraining techniques , are out of the scope of this work and not investigated. We understand it's resource-intensive to conduct experiments at such scale. However this is the way to extract meaningful data points and observation to drive reliable conclusion and guide design of LLM training.

**Reviewer Scores:**

I would expect the reviewers may slightly increase scores during the rebuttal with the additional experiments and discussion. However, with the concerns discussed above in details about scale of experiments and impact of findings, there is no clear distinction that would likely convince reviewers to accept this paper.

---

### Decision · Program_Chairs · 2026-01-26

Reject